



# On the semi-annual variation of relativistic electrons in the outer radiation belt

Christos Katsavrias[1,2], Constantinos Papadimitriou[1,2], Sigiava Aminalragia-Giamini[1,2], Ioannis
A. Daglis[1,3], Ingmar Sandberg[2], and Piers Jiggens[4]

[1]Department of Physics, National and Kapodistrian University of Athens, Greece
[2]Space Applications and Research Consultancy (SPARC), Athens, Greece
[3]Hellenic Space Center, Athens, Greece
[4]ESA/ESTEC, Netherlands

**Correspondence:** Christos Katsavrias (ckatsavrias@phys.uoa.gr)

**Abstract.** The nature of the semi-annual variation in the relativistic electron fluxes in the Earth's outer radiation belt is investigated using Van Allen Probes (MagEIS and REPT) and GOES (EPS) data during solar cycle 24. We perform wavelet and cross-wavelet analysis in a broad energy and spatial range of electron fluxes and examine their phase relationship with the axial, equinoctial and Russell-McPherron mechanisms. It is found that the semi-annual variation in the relativistic electron fluxes exhibits pronounced power in the 0.3–4.2 MeV energy range at L-shells higher than 3.5 and, moreover, it exhibits an in-phase relationship with the Russell-McPherron effect indicating the former is primarily driven by the latter. Furthermore, the analysis of the past 3 solar cycles with GOES/EPS indicates that the semi-annual variation at geosynchronous orbit is evident during the descending phases and coincides with periods of a higher (lower) HSS (ICME) occurrence.

## 1 Introduction

The outer radiation belt consists of electrons with energies from hundreds of keV to several MeV and its response to geospace disturbances is extremely variable spanning from a few hours to several days or even months. Concerning the short-term (a few hours to a few days) variability, previous studies have shown that the trapped relativistic electron population, in the near-Earth space, can be enhanced, depleted, or even not affected at all due to the interplay of acceleration and loss mechanisms (Reeves et al. , 2003; Turner et al. , 2015; Reeves and Daglis , 2016; Katsavrias et al. , 2019a; Daglis et al. , 2019).

Nevertheless, the relativistic electron fluxes in the outer radiation belt also show variations on longer time scales exhibiting a semi-annual as well as an annual periodicity. Even though this semi-annual variation (henceforward SAV) has long been recognized in geomagnetic activity (Cortie , 1912; Chapman and Bartels , 1940), in the radiation belt electron fluxes it was reported for the first time in Baker et al. (1999) using 2 – 6 MeV electron flux measurements from SAMPEX satellite. Over





the past 20 years, the cause of the SAV in the relativistic electrons of the outer belt is still under debate and three possible mechanisms have been proposed:

1. the axial effect (Svalgaard, 1977); the variation of the position of the Earth in heliographic latitude ($\lambda$) resulting to a varying exposure of the terrestrial magnetosphere to high speed solar wind streams (e.g. coronal holes),

2. the equinoctial effect (Cliver et al., 2000; 2002); that is the varying angle of the Earth's dipole ($\psi$) with respect to the Earth-Sun line (and consequently the solar wind speed) with the angle being at 90 degrees during the equinoxes, and

3. the Russell-McPherron effect (Russell and McPherron , 1973), an effect due to the larger z-component of the interplanetary magnetic field (IMF) near the equinoxes in GSM coordinates which results from the tilt of the dipole axis with respect to the heliographic equatorial plane ($\theta$).

The daily values of these three angles are plotted in Figure 1 in a similar way with Cliver et al. (2002).

Since Baker et al. (1999) several studies have attempted to shed light into the cause of the SAV occurrence. Li et al. (2004) used 8 years of SAMPEX electron flux measurements in the 2–6 MeV energy range and divided the SAV into two parts: a semi-annual variation due to the response of the magnetosphere to the solar wind, such as the equinoctial effect, and a semi-annual variation in the solar wind itself in GSM coordinates, such as the axial and Russell-McPherron effects. They argued that the semi-annual variation of the Dst index and MeV electrons deep in the inner magnetosphere can be attributed mostly to the equinoctial effect (orientation of the Earth's dipole axis relative to solar wind flow) with the axial (heliographic latitude) and the Russell-McPherron (IMF z-component in GSM coordinate) effects also contributing while the semi-annual variation of MeV electrons at geostationary orbit is attributed mostly to the semi-annual variation of solar wind velocity as seen by Earth. A few years later, Kanekal et al. (2010) argued that while the equinoctial mechanism may be the dominant mechanism for the seasonal dependence of the geomagnetic activity, the southward component of the IMF plays a crucial role in determining which geomagnetic storms result in increased electron fluxes and which do not, therefore this may account for the dominance of the Russell-McPherron effect as far as relativistic electrons are concerned.

Furthermore, Emery et al. (2011) used >2 MeV electron flux from GOES and argued that the semi-annual periodicity, which was relatively strong in the 1995 – 1997 solar minimum, was a combination of the Russell-McPherron effect and the appearance of equinoctial peaks in the amplitudes of solar rotation periods of 13.5 and 27 days. Finally, Poblet et al. (2020) used GOES >2 MeV electron fluxes during the 1987 – 2008 time-period and concluded that the equinoctial mechanism seems to be the dominant driver of the SAV of electron fluence at GEO.

This study aims in the investigation of the causes of this semi-annual variation by exploiting the high resolution data of the Van Allen probes and to the estimation of its occurrence – during Solar cycle 24 (henceforward SC24) – using sophisticated wavelet techniques (e.g. cross-wavelet and wavelet coherence).

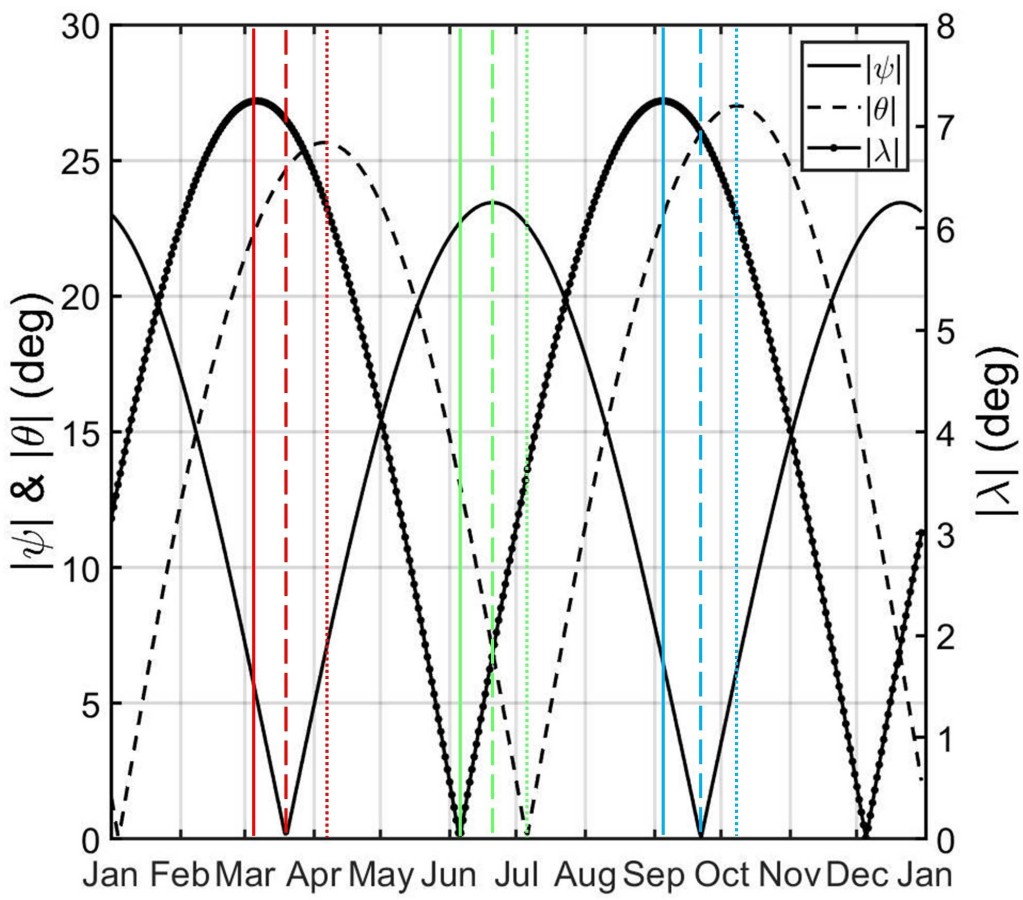

**Figure 1.** Annual profiles of the absolute value of: the $\lambda$ angle governing the axial mechanism (dotted black line), the solar declination $\psi$ governing the equinoctial mechanism (solid black line), and the $\theta$ angle responsible for the Russell-McPherron effect (dashed black line). The vertical lines correspond to the equinoxes maxima (red and blue) and the summer solstice minimum (green) derived from the three mechanisms (Cliver et al. , 2002).

## 2 Data and Methods

### 2.1 Data selection and pre-processing

We use the spin-averaged differential fluxes from the Magnetic Electron Ion Spectrometer (Blake et al. , 2013) and the Relativistic Electron Proton Telescope, REPT, (Baker et al. , 2012) on board the Radiation Belt Storm Probes (RBSP). The dataset spans the time period from January 2013 up to July 2019, which corresponds to the late maximum and descending phase of Solar cycle 24.

Over the early part of the mission, the MagEIS instruments underwent several major changes to energy channel definitions, operational modes, and flux conversion factors. Therefore, in this study, we will focus on data from September 2013 onward,





**Table 1.** MagEIS and REPT energy channels used in this study.

| Instrument | Energy (MeV) |
|---|---|
| MagEIS | 0.033 ; 0.054 ; 0.080 ; 0.108 ; 0.143 ; 0.184 ; 0.235 |
|  | 0.346 ; 0.470 ; 0.597 ; 0.749 ; 0.909 ; 1.575 ; 1.728 |
| REPT | 1.8 ; 2.1 ; 2.6 ; 3.4 ; 4.2 ; 5.3 ; 6.3 |

when the major operational changes were mostly complete. We use the background corrected data (Level 2–see also Claude-pierre et al. (2015) using measurements where the background correction error is less than 75%, only similar to Boyd et al.

(2019). From this procedure, several energy channels from both MagEIS units are excluded due insufficient amount of data. REPT channels, especially those with higher energies (> 5 MeV), are often dominated by background measurements induced by contamination due to galactic cosmic rays. This results in a flattening of the spectrum at channels 6–12 (E > 5.3 MeV). For the lower energy channels, the foreground signal is always much stronger than the background, so no correction is needed. Background is extracted by applying a sinusoidal fit in the flux data at L>6 (GEO) following Boyd et al. (2019). Table 1 shows

the nominal energy values of the combined RBSP A and B channels used. The L-shell values are obtained from the magnetic ephemeris files of the ECT Suite (https://www.rbsp-ect.lanl.gov/science/DataDirectories.php/), which are calculated using the Tsyganenko and Sitnov (2005) magnetospheric field model (TS05).

We have also analyzed electron integral flux measurements with energies > 2 MeV from the Energetic Particle Sensor (EPS) on-board NOAA GOES satellites (https://satdat.ngdc.noaa.gov/sem/goes/data/), starting with GOES-07 in January 1993 and

extending to July 2019 through GOES-15 (Onsager et al. , 1996).

For the calculation of the angles $\lambda$, $\psi$ and $\theta$ governing the axial, the equinoctial and the RM effect, we used the International Radiation Belt Environment Modelling (IRBEM) library (Bourdarie and O'Brien , 2019).

For the performed spectrum analysis, we have used the electron fluxes which initially were found at near-equatorial pitch angles ($a_{eq}$>75 deg). This was done in order to restrict the investigation to measurements of near-equatorial mirroring electrons

which correspond to the majority of the population and, moreover, are less affected by pitch angle scattering effects (Usanova et al. , 2014).

### 2.2 Methods

In this study, following Katsavrias et al. (2016), we make use of the Continuous Wavelet Transform, the Cross-Wavelet Transform and the Wavelet Coherence.





### 2.2.1 Continuous wavelet


The analysis of a function in time, be it F(t), into an orthonormal basis of wavelets is conceptually similar to the Fourier Transform. However, Fourier is only localised in frequency while the Continuous Wavelet transform (hence forward CWT), being localised in frequency and time, allows the local decomposition of Non-stationary time series providing a compact, two dimensional representation (Torrence and Compo , 1998). The wavelets forming the basis are derived from an integrable zero-mean mother wavelet $\psi$(t) and the wavelet transform of F(t), be it W(t, f), is calculated as the convolution of this function with the mother wavelet appropriately shifted and scaled in time:


$$W(t,f) = \int_{-\infty}^{\infty} F(\tau)\sqrt{f}\psi^*[f(\tau-t)]d\tau \tag{1}$$

As mother wavelet we use the Morlet wavelet, which is the commonest function used in astrophysical signals expansions; this allows for a straightforward comparison with previously published work. Furthermore, due to its Gaussian support, the Morlet wavelet expansion inherits optimality as regards the uncertainty principle (Morlet et al. , 1983). Along with the wavelet power spectrum, the global wavelet spectrum is also used which corresponds to the average of the wavelet power spectral density in a specific frequency (f):


$$\overline{W(f)} = \frac{1}{N}\sum_{n=1}^{N} \|W_n(f)\| \tag{2}$$

The global wavelet spectrum generally exhibits similar features (and shape) as the corresponding Fourier spectrum.

### 2.2.2 Cross-wavelet transform


The Cross Wavelet Transform (hence forward XWT–see also Grinsted et al. (2004) between two time-series X and Y and their corresponding CWTs is defined as:

$$W_n^{XY}(f) = W_n^X(f) \cdot W_n^Y(f)^* \tag{3}$$

The result is, in general, complex; the phase relationship between the two variables is then defined as:


$$\Phi = tan^{-1}\left[\frac{im(|W_n^{XY}(f)|)}{re(|W_n^{XY}(f)|)}\right] \tag{4}$$

As shown, the XWT examines the causal relationship in time frequency space between two time series searching for regions of high common power and consistent phase relationship.





### 2.2.3 Wavelet coherence

The wavelet coherence (hence forward WTC) is an estimator of the confidence level for each detection of a time–space region

of consistent phase relationship even if the common power is low. The measure of wavelet coherence closely resembles a localized correlation coefficient in time–frequency space and varies between 0 and 1. It is used alongside the XWT as the latter appears to be unsuitable for significance testing the interrelation between two processes (Maraun and Kurths , 2004). The statistical significance level of the WTC is estimated using Monte Carlo methods (see also Grinsted et al. (2004)).

## 3 Results and discussion

### 3.1 Observations

Figure 2 presents the results of the superposed epoch analysis for the 0.346, 0.749, 1.575, 2.6, 3.4 and 4.2 MeV electron fluxes from MagEIS and REPT during the 2013–2019 time-period which corresponds to the late maximum and descending phase of SC24. The zero epoch time in each plot corresponds to the first day of the year (the extra day corresponding to leap years was not used as it is expected to have negligible effect on the results). The data shown are daily averages, further smoothed using a

28-day moving average window in order to remove any effects of the Solar rotation, while the vertical lines correspond to the equinoxes maxima and the summer solstice minimum derived by the three mechanisms (Cliver et al. , 2002).

It is evident that there are two distinct islets (peaks in relativistic electron flux) in all energy channels, centred roughly on $4<L<5$ (with the exception of 0.346 MeV which spans the $4<L<6$ L range). In detail, the first peak occurs during May (approximately one, one and a half and two months after the RM, equinoctial and axial maxima, respectively), while the second one

occurs almost simultaneously with the RM maximum and lags ≈18 and ≈30 days the equinoctial and axial maxima, respectively. Moreover, these peaks are accompanied by secondary maxima which occur almost simultaneously with the equinoctial and axial maxima, respectively. This behaviour has been observed before (see also Kanekal et al. (2010) and figure 2 therein). The latter authors using, 10 years of SAMPEX data (1993–2002), demonstrated that this asymmetry between the lags of the spring and autumn equinox existed in both the descending phase of SC22 and the ascending phase of SC23, with the latter

being even more prominent than the former. They further suggested that this asymmetry is either due to the limited dataset they used or due to the different ways that high speed streams and coronal mass ejections energize relativistic electrons. We must note here that there can be no straightforward comparison between SAMPEX (low earth orbit and a broad range of equatorial pitch angles) and the dataset considered in this study (near-equatorial elliptical orbit and near-equatorial pitch angles only). Nevertheless, our results indicate that this asymmetry is equally prominent during the descending phase of SC24.

Figure 3 shows the superposed epoch analysis using integral flux measurements of > 2 MeV electrons from GOES/EPS. Note that the longer duration of available data from GOES/EPS allows us to compare the different SC phases, thus, we plot the superposed epoch during three different time-periods: the whole dataset (solid blue line), the ascending phase and maximum (solid red line) and the descending phase (solid green line). As shown, the aforementioned asymmetry is exhibited at geostationary orbit as well. Concerning the whole dataset (blue line), once again the first peak occurs during May, while the second



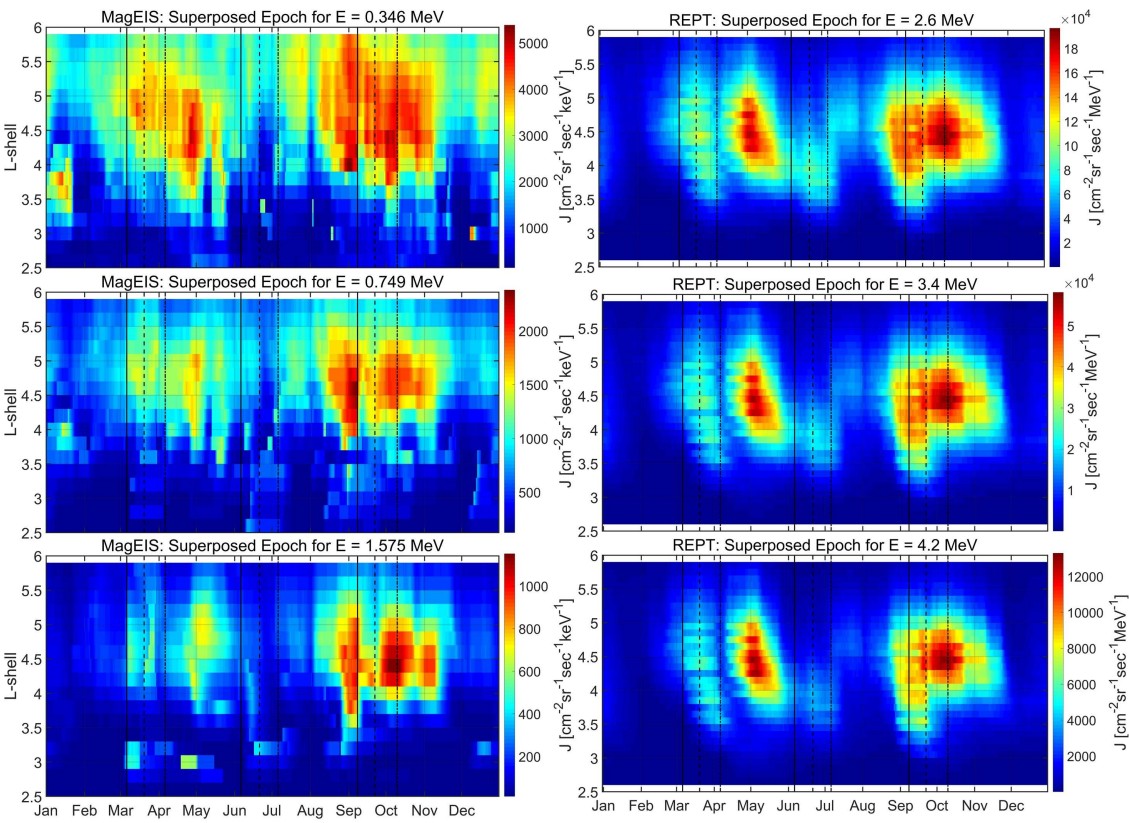

**Figure 2.** Annual superposed epoch analysis showing 28-day moving average of electron fluxes (2013–2019) during the late maximum and descending phase of SC24. Left panels correspond to MagEIS channels (top to bottom: 0.346, 0.749 and 1.575 MeV) and right panels to REPT channels (top to bottom: 2.6, 3.4 and 4.2 MeV). Similar to Figure 1, the vertical lines correspond to the equinoxes maxima and the summer solstice minimum predicted by the three mechanisms (Cliver et al. , 2002).

one occurs simultaneously with the RM predicted maximum, during early October. Similar behaviour is exhibited concerning the secondary maxima. The flux during the descending phase (green line) exhibits the same behaviour. On the other hand, the flux during the ascending and maximum phase (red line) exhibits a completely different behaviour. It increases up to a first peak which occurs between the equinoctial and RM predicted maxima and then, forms a plateau with small variations up to July where the predicted minimum of the RM hypothesis occurs. Then it decreases up to late September (predicted maximum

of the equinoctial mechanism) and forms a shorter-lived maximum during October. The evolution of the superposed flux during the ascending phase and maximum of SC24 indicates that there is not only an asymmetry between the SC phases but the SAV has almost disappeared.

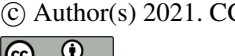



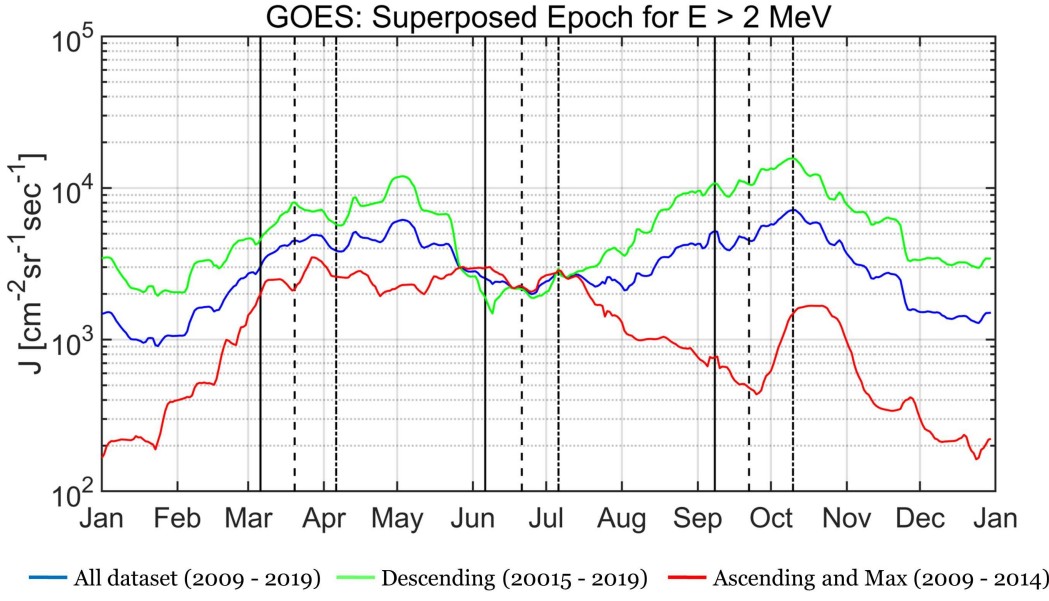

**Figure 3.** Annual superposed epoch analysis showing 28-day moving average electron fluxes at E>2 MeV from GOES/EPS during three time-periods: the whole SC24 (solid blue line), the ascending phase and maximum (solid red line) and the descending phase (solid green line). Similar to Figure 1, the vertical lines correspond to the equinoxes maxima and the summer solstice minimum predicted by the three mechanisms (Cliver et al. , 2002).

## 3.2 The Semi-annual variation (SAV)

Figure 4 presents the time-series, wavelet spectra (CWT) and global wavelet of the 0.346, 0.749, 1.575, 2.6, 3.4 and 4.2 MeV electron fluxes at 4.5<L<5 during the 2013 – 2019 time-period. The black and red solid lines in the time-series correspond to the daily values and to a 28-days moving average, respectively.

In order to reveal the SAV in the corresponding time-series by eliminating the pronounced solar rotation, we have applied a low-pass inverse Chebyshev filter (Williams and Taylors , 1988) with a cut-off period at 33 days; thus, the CWTs (and consequently the global spectra) are calculated using the aforementioned filtered time-series. The filtered time-series exhibit specific bands of periodic behaviour with specific duration which is defined by the 95% confidence level (black contours in the CWT spectra). Moreover, the global spectrum–which resembles a Fourier spectrogram–shows the frequency range of each periodic band along with its maximum, while the dashed red line corresponds to the 95% confidence level. We must note here that the 95% confidence level in the CWT and the global spectrum has different meaning even though they are both sample dependent. The former indicates whether a specific variation is statistically significant (or not) for a finite time-period of the sample, while the latter indicates whether a specific variation is statistically significant for the entire sample.

As shown in the CWT and the global wavelet spectra of the corresponding time-series there is a pronounced SAV (maximum at ≈175 days) at all energy channels which spans the time-period 2015–2018, while it is completely absent during the late

**Figure 4.** Time-series, Wavelet power (CWT) and global wavelet spectra of: (a) 0.346, (b) 0.749, (c) 1.575, (d) 2.6, (e) 3.4 and (f) 4.2 MeV electron fluxes at 4.5<L<5; the red lines are a 28-days moving average smoothing of the time-series. The Wavelet power display is colour-coded with yellow corresponding to the maxima; the black line is the cone of influence of the spectra, where edge effects in the processing become important, while the black contours correspond to the 95% confidence level. The dashed red lines in the global spectra represent the 95% confidence level of the global power.

maximum of the SC24 (2013–2014). Note, that the SAV is present in the descending phase of SC24 at all energy channels (E>100 keV) and at L>3.5, while at L<3.5 is mostly below the 95% confidence level.

Figure 5 shows the CWT of the > 2 MeV electron flux from GOES/EPS during the whole SC24 (2009–2019). As shown, the SAV is exhibited at geostationary orbit, once again, with pronounced power (above the 95% confidence level) during the descending phase of SC24, while it's absent from any other time-period in the dataset. This behaviour is in agreement with the results shown in Figure 3. As mentioned before, Kanekal et al. (2010) argued that a possible explanation for the observed asymmetry between ascending and descending SC phase is the different way that high speed streams and coronal mass ejections

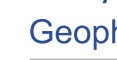
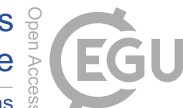

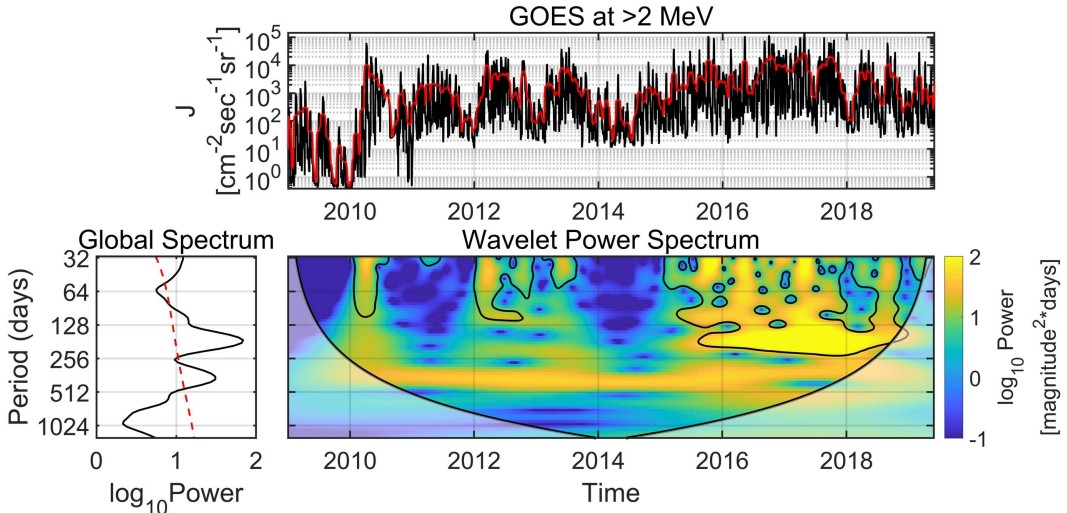

**Figure 5.** Same as Figure 4 for the >2 MeV electron fluxes from GOES/EPS during the whole SC24 from 2009 until 2019.

energize relativistic electrons. Grandin et al. (2019) studied the solar wind high-speed streams (HSSs) emanating from coronal
holes during 1995–2017, encompassing three descending phases (SC22, 23 and 24). In their work they showed the number of
occurrence of Interplanetary Coronal Mass Ejections (ICMEs) and HSSs with solar wind speed larger than 500 km/s.

In order to investigate the aforementioned scenario we compared the CWT of >2 MeV electron flux from GOES/EPS with
the number of occurrence of ICMEs and HSSs during 1993–2019 (taken from figures 3 and 5 in Grandin et al. (2019)). As
shown in Figure 6, the SAV occurs during all three descending phases, roughly during 1994–1996, 2004–2007 and 2015–2018.
The common feature between the occurrences of the SAV is the simultaneous increase (decrease) of HSS (ICMEs), with the
exception of 2004 and 2005 where both are increased. These results indicate that the SAV in the relativistic electrons at GEO
is not only a manifestation of the different reconnection rates produced by the equinoctial/RM mechanisms but a combination
of the latter with the simultaneous occurrence (absence) of HSSs (ICMEs). We must note here that during periods of high
occurrence of both ICMEs and HSSs, the effect of coherent magnetic structure such as an ICME (or at least its magnetic cloud
which can cause long-lasting southward IMF and, thus, long-lasting reconnection) is far more prominent than the modulation
of reconnection produced by the variability of the controlling angles of the equinoctial/RM mechanisms.

### 3.3 Common periodic behaviour and phase relationship

In order to investigate the effect of each mechanism (axial, equinoctial and RM) on the generation of the SAV in the relativistic
electron fluxes we use specific functions of the aforementioned angles instead of the angles themselves. For the $\psi$ angle, which
controls the equinoctial mechanism we use the Svalgaard (1977) function: $S = 1.157 \cdot [1 + 3 \cdot cos(90^o - \psi)^2]^{-2/3}$. Moreover,
similar to Akasofu (1981), we use the $\theta$ and $\lambda$ angles, which control the RM and the axial mechanism, respectively, as $sin(\theta/2)4$
and $sin(\lambda/2)4$. Figure 7 shows the cross-wavelet transform (XWT) and wavelet coherence (WTC) calculations used to study

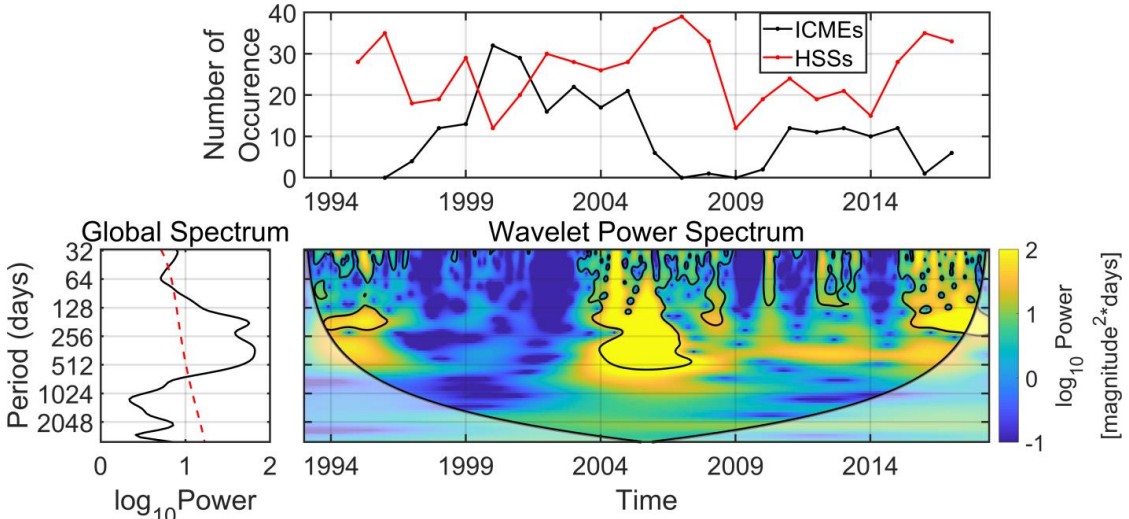

**Figure 6.** Comparison of the occurrence of the SAV in the >2 MeV electron flux from GOES/EPS and the number of occurrence of ICMEs and HSSs during the 1993 – 2019 time-period.

the interrelation of the >2 MeV electron flux from GOES/EPS and the three parameters at geostationary orbit during the whole

SC24. The middle panels show the XWT spectrum of the two time-series under examination. The left panels depict the time-average of the XWT spectrum, which once again resembles a Fourier periodogram, and the right panels the WTC. The latter is the correlation coefficient of the time-series wavelet transform phase. Arrows indicate the phase relationship between the two time-series in time–frequency space: those pointing to the right correspond to in-phase behaviour those to the left anti-phase. The downwards pointing arrows indicate 90 degrees lead of the first data-set.

As shown, the SAV is shared between the >2 MeV electron flux and all three parameters with a maximum power at ≈175 days (left panels of Figure 7). In detail, the SAV in all panels exhibit the maximum power during the descending phase of SC24. Moreover, it appears with reduced (but still significant) power during 2012–2013 and 2009–2010. Note, that during 2014, where the SAV fades, we have the maximum activity of SC24 in terms of the Solar Flare Index provided by the National Oceanic and Atmospheric Administration (NOAA - see also https://www.ngdc.noaa.gov/stp/space-weather/solar-data/solar-

features/solar-flares/index/).

The phase relationship between the flux and the three functions is significant during the descending phase of SC24 (2015–2018), only; a time-period during which the maximum power of the cross spectrum occurs, as well. The important difference in the cross spectra between the flux and the three functions lies on the phase relationship. As shown in both XWT and WTC, the $\theta$ function of the RM effect is the only one in-phase with the electron flux at GEO during the whole descending phase (arrows

continuously pointing to the right). On the other hand, the $\lambda$ exhibits a ≈80–90 degrees phase which corresponds to ≈39–44 days' time-lag, while the S function exhibits a ≈30 degrees phase which correspond to ≈15 days' time-lag). These time-lags and the phase relationships are in agreement with the results presented in Figure 3 concerning the second maximum near the





autumn equinox. They further indicate that the observed SAV at the >2 MeV electron flux at GEO during the descending phase of SC24 is primarily driven by the RM mechanism.

Figure 8 shows the phase relationship between relativistic electron fluxes (0.346 and 0.749 MeV) from MagEIS and the three angle functions as inferred from the XWT spectrum as a function of L-shell and time. Phase is color-coded with green corresponding to zero degrees. The black contours correspond to the coherence level estimator as inferred from WTC. Note that we only show data at L>3.5 since the SAV in both the CWT and the XWT is below the 95% confidence level at L<3.5.

As shown, the results in the outer radiation belt are in agreement with the results at GEO (see also Figure 7). In both energy

channels, the SAV exhibits an in-phase relationship with the $\theta$ function, which controls the RM effect. We emphasize the fact that the deviations in phase are within the 0–10 degrees range, corresponding to a maximum time-lag of 5 days. This behaviour is consistent at the 4<L<5.5 range during the late 2014–2018 time-period with coherence levels >0.6. At L>5.5 the in-phase relationship between flux and the $\theta$ function is limited in the 2016–2018 time-period. On the other hand, the phase relationship between flux and S ($\lambda$) function is mostly $\approx$30 ($\approx$90) degrees, which, as mentioned before, corresponds to a $\approx$15 (45) days'

time-lag. As we move to higher electron energy (Figure 9), there are small deviations from the aforementioned pattern. At 1.575 and 4.2 MeV the flux is in-phase with the $\theta$ function with coherence levels exceeding the 0.6 level mostly during the 2015–2018 time-period and at L>4.5. As in the lower energy channels, the phase relationship between 1.575 and 4.2 MeV electron flux and S ($\lambda$) function is mostly $\approx$30 ($\approx$90) degrees.

### 3.4   Discussion

These results verify that the SAV in the relativistic electron fluxes of the outer radiation belt is primarily driven by the $\theta$ angle which controls the RM effect and, moreover, is present during the descending phase of SC24 (2015–2019). Nevertheless, we cannot exclude some contribution from the equinoctial mechanism as well, since the variation of the phase between electron fluxes and the S function can reach $\approx$15 degrees which corresponds to $\approx$7.5 days' time-lag. The derived results are in agreement with Kanekal et al. (2010) who argued that the times of peak fluxes of relativistic electrons lag the nominal equinoxes

significantly and, therefore, the equinoctial mechanism cannot account for the observed SAV as previously suggested by Li et al. (2004). The fact that these results are consistent through multiple SCs and with various in-situ data renders the conclusions significantly important.

Moreover, as shown in the previous sections, the presence of the SAV in the relativistic electrons of the outer belt coincides with enhanced HSSs occurrence during the descending phase of the SC. This was also indicated by Baker et al. (1999). The

latter authors had proposed the following scenario: the HSSs produce substorm injections due to the effective southward IMF component which, in turn, is favoured by the RM effect (and other factors). Then the injected source/seed populations of low-energy electrons into the inner magnetosphere are accelerated by ULF waves produced by the Kelvin–Helmholtz instability which is also produced by the HSSs. McPherron et al. (2009) provided further evidence on the validity of the aforementioned scenario highlighting the importance of the azimuthal electric field (Ey). On the other hand, Kanekal et al. (2010) showed that

radial diffusion could not explain the simultaneous and rapid flux peaks over a broad range of L-shells and proposed that, even





though the HSSs were responsible for the elevated fluxes during the equinoxes of the descending phase, in-situ acceleration rather than radial transport process may dominate electron energization.

Regardless of the acceleration process, all aforementioned authors agree on the dependence of SAV (and consequently of the electron flux increase around the equinoxes) on the combination of HSSs and the RM effect. Nevertheless, we must note

that recent studies have shown that HSSs are equally (or more) effective in enhancing ultra–relativistic electrons than a major geomagnetic storm produced by an ICME (Horne et al. , 2018; Katsavrias et al. , 2019b).

Finally, there is still an open question left concerning the observed asymmetry between the spring and autumn equinox which cannot be explained by the phase variation inferred from this study.

## 4   Conclusions

In this work we have exploited a broad energy range dataset ($\approx$0.3–4.2 MeV) provided by the MagEIS and REPT instruments on board RBSP in order to investigate not only the occurrences but also the drivers of the SAV of the relativistic electron fluxes in the outer radiation belt.

Our results indicate that the SAV in the relativistic electron fluxes at both GEO and the outer radiation belt (L>4) is exhibited during the descending phase SC24, roughly spanning the 2015–2018 time-period. In order to investigate the consistency

of this result during different SCs we used the >2 MeV integral electron fluxes derived from the Energetic Particle Sensor (EPS) on board the geostationary GOES satellites, covering almost three SCs from January 1993 to July 2019. The wavelet spectrum showed that the SAV occurred during all three descending phases and, moreover, coexisted with periods of increased (decreased) number of HSS (ICME) occurrence indicating that the SAV is a result of the modulation of reconnection produced by the variability of the controlling angles of the RM (and/or equinoctial) mechanism during periods of enhanced solar wind

speed. Unfortunately, this conclusion can be verified only at GEO since RBSP data cover less than a full SC.

Furthermore, we applied the cross wavelet and wavelet coherence techniques in order to investigate the phase relationship between the relativistic electron fluxes and the controlling angles of the axial, equinoctial and RM mechanisms. Our results indicate that the SAV in the relativistic electrons of the outer radiation belt is primarily driven by the RM effect without excluding small contribution from the equinoctial mechanism.

The aforementioned results can be used to refine on-going developments or further contribute to the radiation belt modelling endeavours. Several specification models of the outer radiation belt are used by the engineering community to design both the orbital characteristics of future missions, as well as the shielding of sensitive instruments on-board. Unfortunately, most of these are either completely static, or include time-variations in an overly simplistic manner. As an example, the standard AE-8 model only comes in two versions for active (AE-8MAX) and quiet (AE-8 MIN) solar conditions (Vette , 1991). The successor

to AE-8, AE-9 (Ginet et al. , 2013) is mostly static, only exhibiting time dependence for specific periodicities (including a 6-month one) in a random fashion, using a Monte-Carlo approach. On the other hand, the ONERA models MEO (Lazaro et al. , 2009) and IGE-2006 (Sicard-Piet et al. , 2008) do exhibit a proper solar cycle dependence, but these models are built using yearly averages and thus cannot account for shorter periodicities. Finally, physics-based models are typically run using as input





the value of a specific geomagnetic activity index, or a larger set of observations of the interplanetary conditions and thus

require accurate predictions of these parameters long into the future. Conversely to all these, incorporating the Semi-Annual variability can be easily performed for any point in time and help produce more realistic outputs from even completely static models.

*Author contributions.* CK drafted and wrote the paper with participation of all coauthors. CP and SAG contributed to software development. IS, IAD and PJ were consulted regarding the data analysis and interpretation of the results.

*Competing interests.* The authors declare that they have no conflict of interest.

*Acknowledgements.* This work has received funding from the European Union's Horizon 2020 research and innovation programme "SafeS-pace" under grant agreement No 281 870437 and from the European Space Agency under the "European Contribution to International Radiation Environment Near Earth (IRENE) Modelling System" activity under ESA Contract No 4000127282/19/NL/IB/gg. The Matlab package of the National Oceanography Centre, Liverpool, UK was used in the calculation of the CWT, XWT and WTC. The authors acknowledge

the RBSP/MagEIS and RBSP/REPT teams for the use of the corresponding data sets which can be found online in https://www.rbsp-ect.lanl.gov/science/DataDirectories.php) and the developers of the International Radiation Belt Environment Modeling (IRBEM) library that was used to calculate the theoretical angles of the axial, equinoctial and Russell-McPherron mechanisms.





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

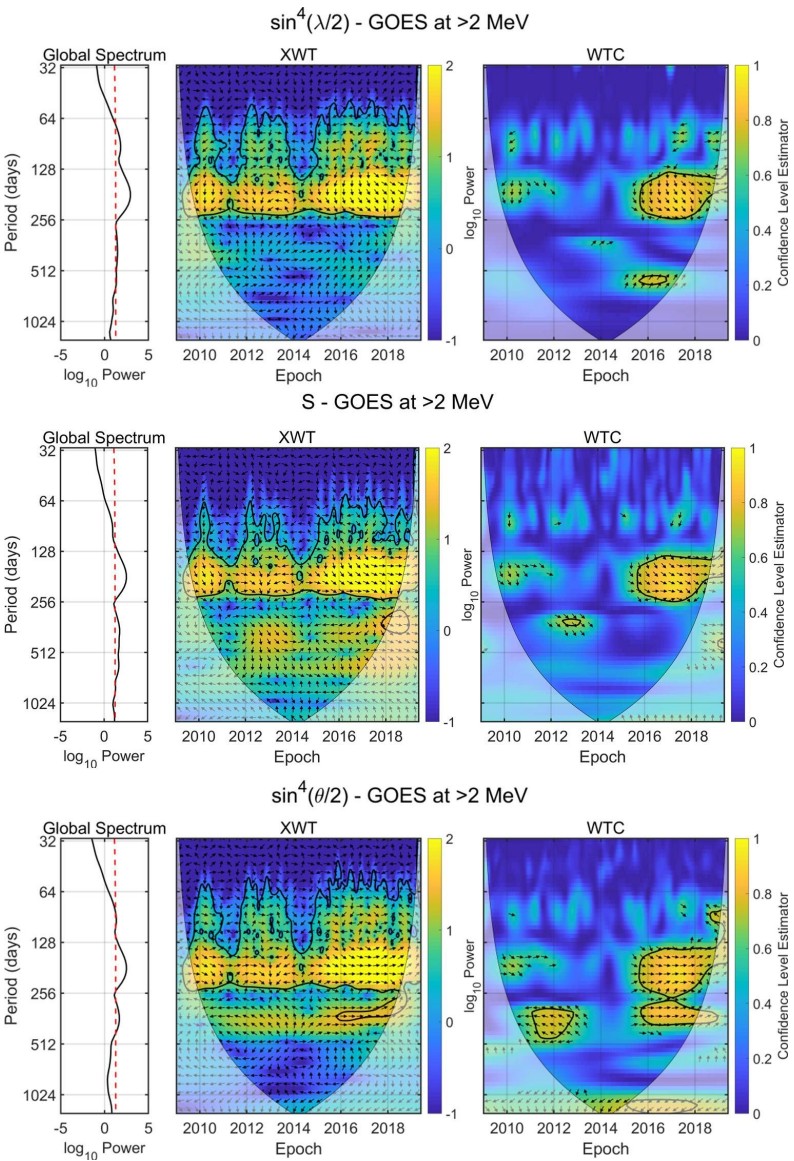

**Figure 7.** Global wavelet (left), cross-wavelet transformation (XWT, middle) and wavelet coherence (WTC, right) between the >2 MeV electron flux from GOES/EPS and: the $\lambda$ function (top panels), the S function (middle panels) and the $\theta$ function (bottom panels); the dashed red line corresponds to the 95% confidence level of the global wavelet. The thick black contours mark the 95% confidence level, and the thin line indicates the cone of influence (COI). The colour-bar of the XWT indicates the log10(power); the colour-bar of the WTC corresponds to the confidence level of the phase obtained by the Monte-Carlo test and the arrows appearing correspond to a confidence level >0.6. The arrows point to the phase relationship of the two data series in time–frequency space: (1) arrows pointing to the right indicate in-phase behaviour; (2) arrows pointing to the left indicate anti-phase behaviour; (3) arrows pointing downward indicate that the first dataset is leading the second by 90 degrees.


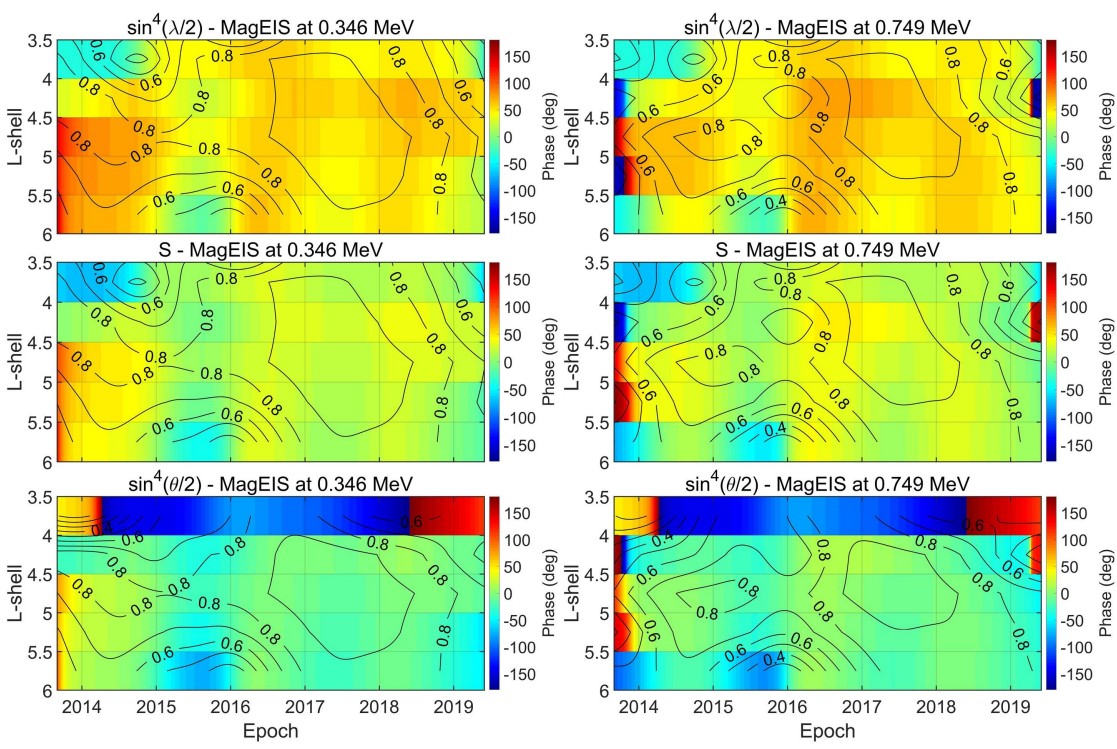

**Figure 8.** Phase relationship between electron fluxes and the three angle functions as inferred from the XWT spectrum as a function of L-shell and time. Left panels correspond to 0.346 and right panels to 0.749 MeV electrons from MagEIS. From top to bottom the phase relationship between flux and $\lambda$, $\psi$ and $\theta$ functions which control the axial, equinoctial and RM mechanism, respectively. The black contours correspond to the coherence level estimator as inferred from WTC.

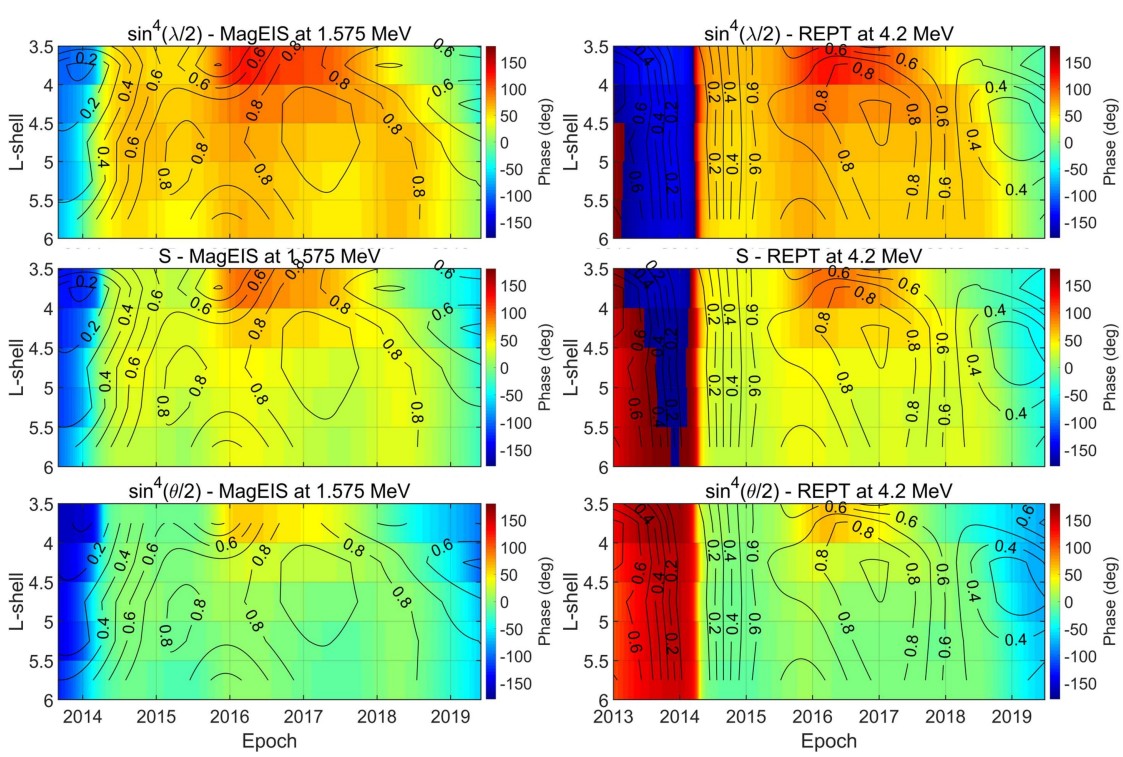

**Figure 9.** Same as Figure 8 for the 1.575 MeV electron flux from MagEIS (left panels) and 4.2 MeV electron flux from REPT.