# Peer review of "On the semi-annual variation of relativistic electrons in the outer radiation belt"

_Annales Geophysicae, 2020_

## Author Response (AR1)

REVIEWER 1

Katsavrias et al. in their work "On the semi-annual variation of relativistic electrons in the outer radiation belt" thoroughly investigate the reasons which produce the variations of the electron intensities. The results can be helpful for the adjustment of the future radiation belt models. The manuscript is well written and structured. I would recommend this study for publication after considering the following minor comments/suggestions:

Specific comments:

lines 59-60: How did you derive that background correction rate was less than 75%? Is that specified in the Level-2 data? What do you mean under "only similar to Boyd et al. (2019)"?

Response: The background correction error is included in the level 2 files of both MagEIS and REPT as FESA_CORR_ERROR. We have only used flux values which have errors less than 75%. The same limit (less than 75%) in the background correction error was also used in Boyd et al. 2019.

line 65, Were both RBSP-A and RBSP-B observations used in this study? How well are the observations from these two spacecraft cross-calibrated?

Response: Yes, we have used both RBSP-A and B measurements. There is no cross-calibration study (to our knowledge so far) between the two spacecraft. Nevertheless, the two spacecraft have identical instruments, which allows assuming that the technical characteristics of the measurements are identical.

line 106, please specify the interpretation of values of 0 and 1.

Response: The text has been amended as follows:

"The measure of wavelet coherence closely resembles a localized correlation coefficient in time–frequency space and varies between 0 and 1, corresponding to non-coherent and highly coherent phase relationship, respectively".

lines 121-122, It is not quite clear what do you mean under "the secondary peaks". Is that the second enhancement after the second islet? And "peaks" mean that those are seen in different energy channels?

Response: We refer to the maxima during March and September, which appear with lower flux values than the primary maxima. The text has been amended as follows:

"Moreover, these peaks are accompanied by secondary maxima (peaks at all energies but with lower flux values than the aforementioned maxima), which occur almost simultaneously with the equinoctial and axial maxima during March and September, respectively."

lines 157-158, about the presence of the significant semi-annual fluctuations for years 2013-2014 and year 2019 we cannot really judge because they are in the cone of influence for this period.

Response: The reviewer is right. We have modified the sentence accordingly:

"Furthermore, the SAV seems to be completely absent during the late maximum of the SC24 (2013--2014) as well as 2019; nevertheless, most of the aforementioned time-period falls inside the cone of influence."

Figure 4, why different time spans are used in panels on the left and right sides?

Response: As we mention in section 2.1, we have excluded all MagEIS data up to September 1st, 2013 due to several major changes to energy channel definitions, operational modes, and flux conversion factors. Thus, even though REPT spans the 01/2013 – 07/2019 time period, MagEIS starts from 09/2013.

line 189, Could you please clarify what does "90 degrees lead of the first data-set" mean?

Response: It means that the first dataset occurs earlier in time and the second follows. The text has been amended accordingly.

lines 200-201, Could you please specify how the phase degrees are transformed in to day's time lag.

Response: The degrees provide us with a fraction of the full cycle which corresponds to the specific period under consideration. For example, let's consider that we have a 90 degrees phase relationship between variables x and y at the semi-annual periodicity. First it means that x is leading y (x precedes y). Furthermore, 90 degrees correspond to a quarter of a full circle and a full circle corresponds to a 6-months period; thus 90 degrees correspond to a quarter of the 6-months, i.e. to 45 days.

line 220, "primarily driven by the theta angle" --> The electron fluxes are not driven by the theta angle. They correlate with the theta angle.

Response: The reviewer is right. The text has been amended as follows: "…primarily driven by the RM effect, which in turn is controlled by the theta angle."

lines 228-241, in study by Smirnov et al., on "Electron Intensity Measurements by the Cluster/RAPID/IES Instrument in Earth's Radiation Belts and Ring Current", JGR, 2019 a clear correlation between AE index, the solar wind dynamic pressure and the electron fluxes at energies of 40 to 400 keV and L-shells 4 to 6 was derived. The enhancement of the solar wind dynamic pressure and AE index occur during the descending phase of the solar cycle. Therefore, these results are also in agreement with those in the manuscript. Namely, they support the scenario of HSS (high speed --> high solar wind dynamic pressure) that drive substorms injections (correlation with AE-index) which populate the radiation belts.

Response: A reference is added in the corresponding paragraph.

"McPherron [2009] provided further evidence on the validity of the aforementioned scenario highlighting the importance of the azimuthal electric field (Ey). Similar results were reported by Smirnov [2019] using Cluster data."

lines 265-272, There is a new model based on the machine learning approach of the electron intensities in the radiation belts at energies 120–600 keV and L-shells ~4 to 7 which is pretty dynamic, depends on the solar wind and geomagnetic conditions and gives high prediction rate: "Medium Energy Electron Flux in Earth's Outer Radiation Belt (MERLIN): A Machine Learning Model", Smirnov et al., 2020, Space Weather. I think it is worse discussing in the outlook. The solar wind electric field does not appear to play highly important role for the electron flux enhancements in this model.

Response: Thank you for this information. The following sentence has been included in the conclusions:

"A new category of models that has emerged in the latest years are machine learning models such as the very recent MERLIN model (Smirnov et al. 2020). These are typically built on many years of data and thus probably include the effects of all such variabilities, but in a way that is difficult to disentangle from all the other effects and variations. Even in these cases though, our study can help in the choice of input parameters, which, when included in a machine learning model, will assist it in properly representing this type of phenomena."

Technical comments:

1. The affiliations 3 and 4 should be added to the affiliation list.

Response: Duly amended.

2. Through the manuscript there is a lot of confusion with definition of abbreviations. They are often defined too late, or several times or not defined at all. For example, HSS(ICME) on line 8 are not defined, fully spelled in line 24, 126, 164 and defined in line 165; MagEIS can be defined either in line 2 or line 52; RM is not defined in line 71; EPS should be defined before line 250.

3. Figure 3, in the horizontal label, "20015"--> "2015"

Response: Duly amended.

4. Please add in line 159 at the end of sentence "(not shown)".

Response: Duly amended.

5. Please acknowledge the data source of GEO data.

Response: Duly amended.

REVIEWER 2

This paper shows new statistical results about the semi-annual variation (SAV) of electron fluxes during solar cycle 24. The authors show that the SAV is mainly explained by the Russel-McPherron effect, and is well correlated with high HSS occurrences. They show that most findings from solar cycles 22 and 23 also apply to solar cycle 24.

These results are of great interest for the radiation belt modelling community. The paper is clear and well written. The figures are mostly clear and appropriate.

I recommend this article for publication, subject to the following minor remarks and questions:

Eq 1,2,3 : some notations are not explained (the star in \phy^* in eq. 1, and what is W_n in eq. 2 and 3). Section 2.2.3 is not very clear, in particular the first sentence. Later in the article the WTC seems to be used as an indicator for the confidence in the XWT phase. It could help to rephrase this section a bit more clearly, and explicit how this metric is used in this study.

Response: ψ is the mother wavelet (here Morlet) and * denotes its conjugate. W_{n}(f)$ is the amplitude of the wavelet at a specific frequency f at the time-stamp with order number n. These are already included in the text. Moreover, we have modified Section 2.2.3 as follows:

"The wavelet coherence (hence forward WTC) is an estimator of the confidence level of consistent phase relationship, between the two time-series, even if the common power is low. The measure of wavelet coherence closely resembles a localized correlation coefficient in time–frequency space and varies between 0 and 1, corresponding to non-coherent and highly coherent phase relationship, respectively. It is used alongside the XWT as the latter appears to be unsuitable for significance testing the interrelation between two processes [Maraun, 2004]. Thus, in our analysis, we are searching for common periodicities which are accompanied by high levels of coherence."

Line 114: How was averaging done (linear or logarithmic average), and how were the data gaps accounted for?

Response: The averaging corresponds to a daily binning and from each bin the linear mean of the fluxes is calculated. For each L-shell bin, in case the data-gaps were less than 30% of the time-series length, we have used a linear interpolation process to fill them. Note that in all cases the maximum consecutive data gaps were 5. In case the data gaps were more than 30% of the time-series length, the corresponding time-series was excluded from further analysis.

Line 140: How statistically significant is this analysis for the 2009-2014 period?

Response: The statistical significance of the 2009-2014 time-period does not differ from the 2015-2019 or the 2009-2019 time-period. All the aforementioned years include measurements from only one GOES satellite (G15) and, moreover, we have followed the previously described procedure for the data-gaps. This means that in every time-period the amount of gaps is less than 30% of the corresponding length and the maximum consecutive data gaps were 5.

Figure 4-7: An horizontal line or indicator at 175 days would help illustrate the discussions. Since all discussions focus on the SAV, why not present only the 175 days horizontal cuts (or a small band around there) of these plots?

Response: A line has been added to guide the reader's eye to the 175 days periodicity.

Line 167: I think there is part of the sentence missing there, do they show that this number correlated with SAV?

Response: We intended to say that these authors displayed, in their work, the number of occurrence of Interplanetary Coronal Mass Ejections (ICMEs) and HSSs with solar wind speed larger than 500 km/s. The sentence is removed to avoid any confusion.

Line 194: "The phase relationship [...] is only significant during the descending phase of SC24". Why is that? Is it because the Wavelet coherence is above 95%?

Response: The reviewer is right. Nevertheless we note that the wavelet coherence (which resembles a correlation coefficient as explained in the first comment's response) takes values between 0 and 1, corresponding to non-coherent and highly coherent phase relationship, respectively.  In the case of the aforementioned figure, the wavelet coherence is higher than 0.7.

Line 271: While SAV could in principle be integrated in specification models, it should be noted that these relatively short-scaled dynamics are only of interest for very specific missions (for instance EOR or short-lived nanosats).

Response: The reviewer is right. A sentence is added at the end of the conclusions section.

EDITOR

Briefly justify the choice to use a linear than a logarithmic interpolation of the particle fluxes or explain if there is any difference in the results when a different interpolation approach is used.

Response: The choice of the linear interpolation was based on simplicity since there is no significant difference between the two interpolation schemes. The following figure shows the filtered time-series of the 0.749 MeV electron flux at L=5.25. The black solid and the red dashed lines correspond to the time-series interpolated using logarithmic and linear scheme, respectively. The corresponding gaps are also shown as black dots (350 in total). As shown, the differences between the two time-series are negligible even if the gaps percentage exceeds 15%.